# Rethinking Vitamin D Deficiency: Controversies and Practical Guidance for Clinical Management

**DOI:** 10.3390/nu17223573

**Published:** 2025-11-15

**Authors:** Manuel Sosa-Henríquez, Óscar Torregrosa-Suau, María Jesús Gómez de Tejada-Romero, María Jesús Cancelo-Hidalgo, Francisco José Tarazona-Santabalbina, Iñigo Etxebarria-Foronda, Guillermo Martínez Díaz-Guerra, Carmen Valdés-Llorca

**Affiliations:** 1D-Evidence Working Group, Spanish Working Group on Vitamin D; otorregrosa@umh.es (Ó.T.-S.); mjgtr@us.es (M.J.G.d.T.-R.); mcanceloh@sego.es (M.J.C.-H.); fjtarazona@gmail.com (F.J.T.-S.); inigo.echevarriaforonda@osakidetza.eus (I.E.-F.); guillermo.martinez@salud.madrid.org (G.M.D.-G.); cvaldesyllorca@gmail.com (C.V.-L.); 2Working Group on Osteoporosis and Bone Metabolic Diseases, Department of Medical and Surgical Sciences, University of Las Palmas de Gran Canaria, 35016 Las Palmas de Gran Canaria, Spain; 3Bone Metabolic Unit, Service of Internal Medicine, Hospital General y Universitario de Elche, 03203 Alicante, Spain; 4Department of Medicine, University of Seville, 41009 Seville, Spain; 5University Hospital of Guadalajara, Alcalá University, 28801 Madrid, Spain; 6Geriatric Service, Hospital University of La Ribera, 46600 Alzira, Spain; 7Department of Orthopaedic and Trauma Surgery, 20014 Gipuzkoa, Spain; 8Instituto de Investigación Sanitaria IMAS 12, Hospital Universitario 12 de Octubre, University Complutense de Madrid, 28040 Madrid, Spain; 9Health Center of Fuencarral, DA North, SERMAS, 28034 Madrid, Spain

**Keywords:** Vitamin D, deficiency, supplementation, clinical screening, calcium homeostasis

## Abstract

Vitamin D (VD), due to its hormonal action, plays a crucial role in calcium homeostasis and bone metabolism, and its deficiency has been associated with musculoskeletal disorders such as osteoporosis, fractures, and osteomalacia, as well as a growing attention of chronic conditions and certain cancers. Despite its physiological relevance and widespread prevalence, particularly among older individuals, patients with chronic diseases, institutionalized populations and pregnant or lactating women, clinical approaches to diagnosing and managing vitamin D deficiency (VDD) remain heterogeneous across guidelines and healthcare settings. This reflects a lack of consensus regarding the benefits and limitations of universal versus selective screening, the definition of adequate serum concentrations, and the clinical indications for supplementation across different patient profiles. This narrative review explores key controversies in the clinical management of VDD, including current perspectives on screening strategies and target populations, indications for empirical supplementation, criteria for biochemical monitoring, and therapeutic goals in bone-related outcomes. In particular, the review discusses the rationale for adopting a 30 ng/mL (75 nmol/L) threshold for adequate serum 25(OH) concentrations in skeletal health, the role of vitamin D and calcium in osteoporosis treatment, and the pharmacological advantages of cholecalciferol compared to other vitamin D compounds. Through a synthesis of available evidence and expert consensus, the review aims to support clinical decision-making in the prevention and treatment of VDD and to identify areas that require further clarification or research. This review aims to support evidence-based clinical decision-making.

## 1. Introduction

Vitamin D (VD), due to its hormonal action, has long been recognized for its essential role in calcium homeostasis and bone health. In recent years, its role has extended far beyond the skeletal system, with increasing evidence supporting its involvement in muscle function, immune modulation, Glucosa homeostasis, cardiovascular function and chronic disease prevention [1]. Yet, despite this growing recognition, vitamin D deficiency (VDD) remains a widespread and under-addressed global issue [2].

The burden of VDD is particularly pronounced among high-risk populations, including older adults, individuals with limited sun exposure due to geographic location, and those with malabsorption disorders [3]. The persistently high prevalence of VDD necessitates clear strategies for the clinically effective, safe, and cost-effective identification, treatment, and monitoring of affected individuals.

Efforts to standardize diagnostic criteria and management strategies have been hindered by a lack of consensus on key aspects of care, including optimal serum 25-hydroxyvitamin D [25(OH)D] thresholds, the appropriateness of routine population screening, and the indications for follow-up testing [4,5]. Since universal screening is controversial due to concerns about cost-effectiveness, empiric supplementation, especially among well-identified high-risk groups, has emerged as a practical and increasingly endorsed alternative. This approach offers a simple and economically efficient means of reducing disease burden without necessitating routine biochemical testing in every case [6].

At the same time, efforts to optimize treatment strategies have reinforced the importance of selecting the most effective form of vitamin D. Among available options, cholecalciferol (vitamin D3) has consistently demonstrated greater potency, a longer half-life, and more stable serum responses than other alternatives, making it the preferred agent in most guidelines and clinical settings [7]. Other alternatives are ergocalciferol (vitamin D2), derived from plant sources; calcifediol (25-hydroxyvitamin D3), which is a pre-hydroxylated form; and calcitriol (1,25-dihydroxyvitamin D3), the hormonally active metabolite, along with analogues such as alfacalcidol and paricalcitol [7,8].

In light of ongoing uncertainties, this narrative review aims to examine the evolving landscape of vitamin D management, address key controversies in screening and supplementation, and provide an evidence-based perspective on optimizing care in both skeletal and extra-skeletal contexts.

### 1.1. Vitamin D Deficiency: Update of Global Prevalence

The high prevalence of VDD remains a critical global public health issue, affecting across age groups, regions, and population subgroups. Although cut-off values remain debated, VDD is commonly defined as serum 25(OH)D concentrations below 50 nmol/L, with severe deficiency often considered below 30 nmol/L [9].

A comprehensive pooled analysis [2], which encompassed nearly 8 million individuals from 81 countries between 2000 and 2022, found that 15.7% of the global population had serum 25(OH)D levels < 30 nmol/L, 47.9% < 50 nmol/L, and 76.6% < 75 nmol/L. These figures remained relatively stable for over two decades showing a slight, but not significant decrease from the 2000–2010 period to 2011–2022. Variations between geographical areas seem to be affected by latitude and socio-economic status which may be due to insufficient sunshine exposition and lack of awareness. The highest prevalence has traditionally been observed in the Middle East and South Asia regions with several countries having over 30% of the population < 30 nmol/L of 25(OH)D levels [2,10,11,12]. In contrast, prevalence in Africa has been usually considered as low due to the ample sunshine or more young living people [2,13,14]. However recent studies have shown that around 18% of the population remains < 30 nmol/L of 25(OH)D levels [15,16]. North America has also displayed low levels of VDD [2], clearly influenced by the prevalence of high-income countries such as United States (2.6% with <25 nmol/L of 25(OH)D [17]) or Canada (7.4% with <30 nmol/L of 25(OH)D [18]) which contrasts with data obtained from South America where the prevalence of 25-(OH)-D levels < 20 ng/mL has been found to be 34.8% [19]. In Europe, despite public health recommendations, the prevalence of serum 25(OH)D levels below 30 nmol/L remains high, estimated at approximately 18% [2], which agrees with a recent cohort study from the United Kingdom Biobank where 13% of individuals were severely vitamin D deficient (25(OH)D values from 10 to 25 nmol/L) [20]. In Spain, the situation for adult population is similar, with 18% of the population with 25(OH)D levels < 30 nmol/L [21].

Seasonal patterns also influence VDD, which is 1.7 times more prevalent during winter-spring than in summer-autumn worldwide [2]. In addition, there are other factors that increase the vulnerability of certain groups, placing them at higher risk of VDD. Age influences VDD risk in both directions; on the one hand, as people grow older, their skin becomes less efficient at synthesizing vitamin D from sunlight, dietary intake often decreases, and mobility limitations may reduce time spent outdoors, all contributing to insufficient 25(OH) concentrations [22]. A recent meta-analysis [23] estimated that approximately 60% of older adults individuals globally have 25(OH) concentrations below 50 nmol/L. On the other hand, many adolescents and children have an urban living and indoor-oriented lifestyle, spending limited time outdoors, especially during colder months, reducing their exposure to sunlight. Several studies have shown high prevalence among young people [2,24,25,26]. Women are also consistently found to have a higher prevalence of VDD [2,27], especially during pregnancy [28,29], breastfeeding [30] or post-menopause [31]. It has long been known that in individuals with pigmented skin, melanin acts as a natural sunscreen and reduces the skin’s ability to produce vitamin D from sunlight exposure [32], which increases the prevalence of VDD among these groups [33,34]. Finally, VDD is highly prevalent in a wide range of chronic disorders, including chronic kidney disease, liver cirrhosis, malabsorptive syndromes such as celiac disease and inflammatory bowel disease, and obesity [35,36].

It is worth noting that some authors have proposed that certain human populations may have undergone physiological adaptations that allow efficient use of vitamin D even at relatively low serum 25(OH)D concentrations. This evolutionary perspective suggests that vitamin D requirements may not be universal across populations. While this hypothesis is of great interest, our recommendations are based on current clinical and public health guidelines that define vitamin D deficiency according to serum 25(OH)D concentrations and their associations with health outcomes in diverse populations.

### 1.2. Vitamin D Testing

Reports have revealed a large increase in the number of laboratory testing for vitamin D over the last several years [37,38]. This is primarily due to the above-mentioned burden of hypovitaminosis D in the global population [39], the link proposed between VDD and several chronic diseases as well as high mortality rates, and the potential role of vitamin D supplementation in improving certain health conditions. Some studies, however, have reported that about 25% to 78% of all these vitamin D tests are inappropriate [38]. The proportion of such unnecessary tests is believed to increase up to 76.5% in high socioeconomic areas [40]. As a result, several countries such as the US and Canada, have tried to adopt various strategies to limit the un-needed vitamin D testing [41,42].

It has been developed more advanced assays, based on liquid chromatography-tandem mass spectrometry (LC-MS/MS), that have become the gold standard for reliable determination of VDD, that allow for the simultaneous measurement of different vitamin D metabolites with high precision and specificity. Additionally, MS methods can distinguish between VD3 and VD2 metabolites, as well as their 3-epi-derivatives, where immunological assays all fail. Particularly, as relevant criteria for functional VDD, the ratio (VMR) of 24,25(OH)2VD to 25(OH)VD (both D2 and D3) have to be considered. VDD means by this 24,25(OH)2VD < 3 ng/mL and VMR < 4% [43].

#### 1.2.1. Universal Screening

Routine screening for VDD remains a prominent subject of debate, with robust scientific evidence cited in support of both positions. Interventions to prevent VDD have been estimated to be highly cost-effective and even cost-saving for some populations because many consider low 25(OH) concentrations to be a contributor to certain diseases rather than only a marker for impaired health [44]. The question, therefore, is whether universal screening is necessary.

It can be argued that population-wide screening could help identify individuals who may benefit from vitamin D supplementation before the onset of adverse clinical outcomes, while minimizing the risk of harm from the intervention and treatment [45,46]. Meta-analyses support this approach, indicating that supplementation in vitamin D deficient older adults can reduce accidental falls [47]. Another potential benefit of screening could be the establishment of a baseline prior to initiating treatment. However, alternative perspectives suggest that measuring 25(OH) concentrations post-treatment may offer greater clinical utility, as it could not only enhance patient adherence but also prove to be more cost-effective [48,49].

To date, no published study has evaluated the effects of a screening program for VDD in the general population, so the evidence is insufficient to balance the benefits and costs of such screening [50,51]. The United States Preventive Service Task Force (USPSTF) commissioned a systematic review on screening for VDD including the benefits and costs of massive screening and early treatment in community-dwelling non-pregnant adults who have no signs or symptoms of VDD or conditions for which vitamin D treatment is recommended. The USPSTF found insufficient evidence to support the benefits of screening for VDD and, consequently, recommended against its implementation [50].

Therefore, given its cost, routine screening for VDD by measuring serum 25(OH)D concentrations in the general population is currently not justified. In line with this, current clinical guidelines do not recommend routine screening, and the American Society for Clinical Pathology explicitly advises against its implementation [51,52,53,54,55,56]. Alternatively, there are currently a number of questionnaires that correctly predict serum 25(OH)D levels in the population as a whole as well as in specific subgroups, including adults [39,57,58,59,60,61].

#### 1.2.2. Population-Based Screening

Consistent with this idea, there is increasing recognition in adopting a more strategic approach to screening, targeting specific populations most likely to benefit from treatment which may lead to improved outcomes [62]. The at-risk groups for VDD include, but are not limited to, individuals with obesity; those with limited ultraviolet radiation exposure (such as residents of assisted living facilities, in areas with scarce sunlight, or even in very sunny areas when wearing full-body coverage); individuals with impaired vitamin D absorption (e.g., due to steatorrhea, bariatric surgery, or other malabsorptive conditions); those with unbalanced diets or increased nutritional requirements (such as vegetarians or during pregnancy); individuals taking medications that alter vitamin D metabolism (e.g., anticonvulsants); and, in particular, those with conditions associated with VDD (such as osteoporosis or osteomalacia) [55,63,64,65,66]. Therefore, vitamin D screening should be performed in at-risk populations, while others should be advised to maintain safe UV exposure, a balanced diet, and regular physical activity.

Indeed, while general screening of 25(OH)D concentrations is not advocated, some professional societies do recommend screening for VDD in individuals at risk, including non-Caucasian individuals, pregnant and lactating women, older adults with a history of falls or non-traumatic fractures, patients with bone diseases, among others. In addition, vitamin D testing may also benefit people with laboratory or clinical findings commonly associated with VDD [53,54,65,67,68]. It is worth mentioning that patients with osteoporosis or those at increased risk of fracture should undergo assessment of their vitamin D status. The rationale for this is that osteomalacia can lead to fractures, and repletion of vitamin D is the treatment of choice in these patients [69]. Secondly, the results of a 24 h urine calcium test, commonly used in the evaluation of osteoporosis, cannot be accurately interpreted unless the patient has an adequate level of vitamin D [70]. Thirdly, most studies on osteoporosis treatment suggest a better response to osteoporosis-specific medications when the patient’s 25(OH)D concentrations is at least 30 ng/mL [71,72]. Finally, antiresorptive drugs used in the treatment of osteoporosis can cause hypocalcemia [73], a risk that is significantly reduced in patients with sufficient vitamin D stores and adequate calcium supplies [74].

However, this population-based screening approach also raises certain concerns. Some experts argue that establishing strict screening criteria may result in certain patients with VDD not receiving appropriate treatment, or may lead to over- or underdiagnosis due to uncertainties surrounding cut-off values and the variability of existing 25(OH)D assays [45,75]. Moreover, not all organizations agree on the definition of at-risk groups, which can be the case for pregnant women [37,76].

Thus, several limitations need to be addressed for a better understanding of the most cost-effective approach at the population level [77]. Total 25(OH)D level is currently considered the best marker of vitamin D status; however, 25(OH)D assays present inherent limitations [78]. Despite ongoing efforts, establishing a clearer definition of VDD and identifying the most cost-effective strategy to detect individuals at risk appear to be the most effective steps to address this issue.

## 2. Empiric/Targeted Vitamin D Supplementation

### 2.1. Summary of Evidence

The well-documented high incidence of hypovitaminosis D in high-risk populations has led to an alternative management strategy: recommending universal supplementation in these groups, emphasizing that measuring 25(OH)D concentrations in such cases may not be cost-effective [39]. In this context, universal vitamin D supplementation—particularly among pregnant and breastfeeding women, as well as young children—has been shown to significantly reduce the incidence of symptomatic VDD [79].

This approach takes into consideration that screening represents a burden on the healthcare system. Although vitamin D testing is relatively less expensive than many other biochemical tests, the high volume of tests, often performed repeatedly, may impose a substantial financial burden [62]. In this sense, the cost of the assay in some countries is up to ten times higher than one year of supplementation [57]. On the other hand, vitamin D treatment is inexpensive and, except in cases involving high-dose regimens, is rarely associated with adverse effects [80]. It should be kept in mind that, although vitamin D toxicity is uncommon, the likelihood of complications may increase when supplementation is extended to larger populations.

#### 2.1.1. Cost-Effectiveness of Population-Based Supplementation

Given the increase in vitamin D testing, national restrictions, as well as controversies between different strategies, the comparison between the cost-effectiveness of 25(OH)D measurement and long-term vitamin D supplementation has garnered increasing attention in recent years.

Among different age groups, the cost-effectiveness of vitamin D supplementation appears most strongly supported in those aged over 70. A study conducted in Ireland assessed the cost-effectiveness of vitamin D3 supplementation in older adults of different age groups (≥50, ≥60 and ≥70 years) with VDD (defined as serum levels below 30 ng/mL). For the ≥70 years age group, the average annual costs and outcomes would be approximately €5.6 million, 1044 QALYs (quality-adjusted life years) gained with a cost/QALY of approximately €5400. The results were most sensitive to changes in the mortality risk reduction attributed vitamin D3 supplementation [81].

In addition, prescription of cholecalciferol 800 IU to older adults aged ≥ 65 years could reduce the incident risk of hip fractures and falls, and prevent associated mortality, potentially yielding millions of pounds in cost savings [82,83]. Comparatively, both population screening and universal supplementation for vitamin D insufficiency have shown to be cost-effective strategies in community-dwelling older women and men [47]. Finally, a study [84] evaluated the cost-effectiveness of calcium/vitamin D supplementation for preventing osteoporotic fractures in individuals ≥65 years old. Altogether, despite that both population screening and vitamin D3 supplementation were similar with regard to cost-effectiveness, 25(OH)D measurement for the evaluation of vitamin D status is not always feasible; therefore, vitamin D3 empiric supplementation would be a noteworthy option in such cases. Additionally, authors coincide that no treatment is consistently the least cost-effective option.

Weaver CM et al. [85] concluded that if all adults with osteoporosis in the US and the European Union (EU) used calcium and vitamin D supplements, it could prevent more than 300,000 fractures/year in the US and more than 500,000/year in the EU and save approximately €5.7 billion and US $3.3 billion annually. Benefits were likely driven by older adults ≥ 65 years. Therefore, calcium and VD would be highly cost-effective. Moreover, among nursing home residents aged ≥ 75 years, a well-known risk group for vitamin D insufficiency, the annual net economic cost of screening all individuals for VDD and supplementing those considered insufficient (25(OH)D levels < 50 nmol/L) was found to be comparable to universal supplementation. This suggests that assessing 25(OH)D concentrations prior to supplementation may not be necessary in the nursing-home residents [75].

These studies assessing the cost-effectiveness of vitamin D supplementation are heterogenous and have different follow-up periods, VDD definitions and even outcome descriptions. The results of those assessing the efficacy of population-based screening, on the other hand, depend greatly on factors such as the cost of vitamin D testing, the reimbursement rate and again the definition of VDD. As a result, the abovementioned findings should be interpreted with care, while determining methods to measure the outcomes of each strategy and designing randomized controlled trials to evaluate clinical outcomes or harm in each group. It can therefore be concluded that more studies are needed to determine a population strategy that is both clinically- and cost-effective and decide whether cost-effectiveness varies by population subgroups and country settings.

#### 2.1.2. Current Guideline Recommendations

Several medical societies and health organizations have already recommended targeted empirical supplementation, in other words, initiating vitamin D supplementation in the absence of testing in different populations [6,86]. The reasons underlying these recommendations are that there is sufficient evidence of subsequent benefits from this vitamin D supplementation and this strategy is generally simple, effective, inexpensive and widely applicable across healthcare settings.

For instance, oral supplements are already recommended when ultraviolet radiation (UVR) exposure is insufficient or undesired and when nutritional sources are scarce based on the Institute of Medicine (IOM) guidelines [87]. The UK Departments of Health and experts recommend vitamin D supplementation to children and adolescents, pregnant and breastfeeding women to reach recommended daily allowances [88,89]. Moreover, the UK government, along with the American Geriatrics Society and the French Group of Geriatrics also indicate that routine laboratory testing for 25(OH)D serum concentrations before supplementation begins is not necessary in older adults (≥65 years old), particularly in those with insufficient sun exposure, such as nursing home residents [49,88,90]. According to the National Osteoporosis Society, routine vitamin D screening may be unnecessary in patients with osteoporosis or fragility fracture, who may be co-prescribed vitamin D supplementation (often with calcium) with an oral antiresorptive treatment [91]. Finally, cholecalciferol was recommended as the preferred compound in the case of treatment in the absence of 25(OH)D determination [66,92].

### 2.2. Recommendations for Vitamin D Empiric Treatment

#### 2.2.1. Population Groups

Therefore, there would be certain categories of individuals for whom supplementation appears justified given the sufficient evidence of subsequent benefits from this strategy (Table 1). These patients may be initially supplemented, with serum 25(OH)D concentrations assessed subsequently. Serum 25(OH)D assay should be requested to confirm attainment of sufficiency, independently of the threshold chosen.

As a brief rationale, empiric vitamin D supplementation without testing can be justified for patients who have no overt risk factors or evidence of deficiency but are thought to have inadequate sun exposure or dietary intake [65]. In pigmented skin people, as the melanin pigment absorbs UVB radiation, it also reduces the UVB radiation available for vitamin D synthesis in the skin and this group is at high-risk of deficiency [93].

Numerous studies show the risk of adverse pregnancy outcomes (e.g., preeclampsia, gestational diabetes mellitus, preterm delivery) can be significantly reduced or even prevented by vitamin D supplementation [98]. Pregnant and breastfeeding women should take a daily supplement to ensure that the mother’s requirements for vitamin D are met and to build adequate fetal stores for early infancy. The vitamin D supplementation should be continued throughout pregnancy and lactation [94].

The production and metabolism of vitamin D change with aging. Causes include decreased sun exposure (particularly institutionalized individuals) and reduced capacity of the skin to produce vitamin D. Therefore, older adults are at a high risk of severe deficiency, and prescribing vitamin D treatment is often necessary. The rationale for supplying adequate amounts is that elevated PTH concentrations may contribute to bone fragility and falls in older adults [95].

Although there continues to be controversy over what is the optimal concentration associated with the greatest suppression of PTH in order to minimize bone resorption, according to published studies, it has been shown that in adults over 65 years of age, the maximum suppression of PTH is achieved with concentrations of 25(OH)D of 72 nmol/L [99]. For this reason, organizations such as the International Osteoporosis Foundation (IOF) have established 75 nmol/L as an optimal cutoff. In extra-bone pathologies, ad hoc studies should be conducted [100].

In osteoporosis patients, VDD may impair the response to osteoporosis treatment [101]. Vitamin D repletion appears to be necessary in order to maximize the response to antiresorptive therapy, as it ensures adequate calcium and PTH levels in the organism. All randomized clinical trials on osteoporosis drugs were conducted under the conditions of vitamin D (cholecalciferol) and calcium supplementation [97,101]. Calcium supplementation is also indicated in patients with osteoporosis and an insufficient dietary calcium intake. Altogether, the effect of antiresorptive drugs may be compromised and there is risk of hypocalcemia in the absence of calcium and vitamin D [89,97]; therefore, their administration before starting osteoporosis treatment with antiresorptive drugs and co-adjuvant therapy is recommended by major osteoporosis guidelines [54,102,103,104].

Current evidence on the role of vitamin D in fracture prevention remains inconclusive, particularly regarding its isolated use. Reported benefits, such as improvements in bone health, reductions in fracture incidence, overall mortality, and fall risk, are primarily observed when vitamin D is administered in combination with calcium [50,75]. However, in the context of secondary prevention, it is essential to correct VDD by achieving serum 25(OH)D levels above 30 ng/mL prior to initiating pharmacological treatment for osteoporosis, as this enhances therapeutic efficacy and reduces the likelihood of hypocalcemia [105]. Vitamin D deficiency is commonly observed in conditions that impair intestinal absorption, including celiac disease, short bowel syndrome, and cystic fibrosis. Additionally, individuals with obesity present a higher risk of deficiency due to the sequestration of vitamin D in adipose tissue, which limits its bioavailability. For this reason, it is crucial to identify and correct VDD both prior to and following bariatric surgery, to reduce the risk of postoperative complications and to mitigate long-term skeletal consequences, such as metabolic bone disease and increased fracture susceptibility. Finally, it must be taken into consideration that functional VDD is a better predictor of morbidity and mortality than just low 25(OH)D concentration.

#### 2.2.2. Therapeutic Strategy

As previously stated by different clinical guidelines and experts, cholecalciferol should be the molecule of choice in the case of treatment in the absence of 25(OH)D determination [66,92]. The following features of oral cholecalciferol response underline this recommendation: Due to its lipophilic and liposoluble characteristics, cholecalciferol has a half-life of approximately 60 days [106], so its administration produces stable, predictable, and sustained plasma levels over time [107,108]. Cholecalciferol has a specific feedback mechanism in its hepatic hydroxylation, which prevents the excessive activity of vitamin D [109], allowing a safe and effective weekly or monthly treatment, in its daily equivalents [110,111]. The administration of 4000 IU/day of cholecalciferol is widely regarded as safe. It is likely to pose no risk of adverse health effects to almost all individuals (Endorsed by the European Food Safety Authority—EFSA) [112].

An initial vitamin D therapy, i.e., loading dose, could be used if a rapid correction of VDD is anticipated or identified. A loading dose of 25,000–50,000 IU/week could be administered for 4–6 weeks, followed by a maintenance dose [113].

Regardless of how vitamin D therapy is initiated, a prevention/maintenance dose would be needed to avoid recurrent deficiency in these populations. Available data indicate that, on average, for each additional 100 IU of vitamin D_3_ ingested daily, the serum 25(OH)D should increase by 1 ng/mL [114,115]. The addition of 2000 IU daily could be expected to increase serum 25(OH)D by 20 ng/mL, an increment with no toxicity [113]. It seems reasonable to recommend a vitamin D supplementation dose of 800 to 2000 international units (IU) per day for adults who want to ensure a sufficient vitamin D status [66].

In older adults, recommended doses for vitamin D vary between 1000 and 2000 IU/day to achieve a goal of circulating 25(OH)D of at least 50 nmol/L (20 ng/mL), but ideally 75 nmol/L (30 ng/mL) should be targeted to maximize bone health [95]. A working group of the European Society for Clinical and Economic Aspects of osteoporosis, osteoarthritis and musculoskeletal diseases (ESCEO) came to the conclusion that 1000 IU/daily should be recommended in patients at increased risk of VDD [116]. Other experts and societies recommend daily doses of up to 2000 IU for this population [54,103,117], especially in patients with osteoporosis, fractured or institutionalized older adults [118]. Special populations, such as patients with malabsorption, obesity, or those who have undergone bariatric surgery, are advised to receive higher doses of vitamin D [66].

Given the persistently low long-term adherence to supplementation, it is important to consider alternative methods of increasing 25(OH)D levels, such as food fortification with vitamin D, biofortification, and the promotion of safe sun exposure practices. These approaches may be particularly valuable in countries where they represent a unique opportunity. Implementing such strategies could reduce both the costs of vitamin D testing and the associated expenses for national healthcare systems, especially in countries where these costs are rapidly increasing.

## 3. Need for Vitamin D Monitoring Following Treatment

Once under treatment, the need for monitoring will be highly influenced by the properties of the form used and dosing regimen [119]. The two most common oral forms of vitamin D are plant-derived ergocalciferol (D_2_) and animal-derived cholecalciferol (D_3_). Although vitamin D_2_ and D_3_ are often considered to be equivalent, there are subtle differences including the affinity of the vitamin D binding protein (VDBP) for their respective metabolites. VDBP has a slightly lower affinity for vitamin D_2_ metabolites compared to those of vitamin D_3_, which likely leads to the approximately 10% shorter half-life of 25(OH)D_2_ relative to 25(OH)D_3_ [119]. Moreover, a systematic review of studies comparing vitamin D_2_ and vitamin D_3_ supplementation found that cholecalciferol produces greater increments in circulating total 25(OH)D concentrations than ergocalciferol [110]. In addition, vitamin D_2_ is not accurately measured in nearly all the 25(OH)D assays. In some parts of the world, intermediate or active metabolites of vitamin D (25(OH)D/calcifediol or 1,25(OH)2D/calcitriol) are available for treatment, particularly for situations of affected vitamin D metabolism, such as patients with severe liver disease or moderate-to-severe renal impairment, 25-hydroxylase and 1-α-hydroxylase deficiency, severe intestinal malabsorption, or hypoparathyroidism [100]. Of note, and generally speaking, these metabolites are not a substitute for adequate vitamin D intake as Vitamin D2–D3 is the substrate for 25(OH)D and the natural pathway by which the body processes vitamin D is synthesizing, storing and metabolizing cholecalciferol. Circulating 25(OH)D concentrations may be important to support the non-renal production of 1,25(OH)2D and in turn, local production of 1,25(OH)2D is important to mediate certain non-skeletal effects of vitamin D [7,100]. Therefore, vitamin D_3_ is the preferred form for clinical use [100,120].

### 3.1. Determinants of Oral Vitamin D_3_ Response

#### 3.1.1. Pharmacokinetics

Pharmacokinetic studies have determined that the serum half-life (t1/2) of cholecalciferol is 2 days, while its functional half-life is 60 days, as its lipophilic and liposoluble nature allows tissue storage. Conversely, 25(OH)D half-life is 15 days; and 1,25(OH)2D is hours in length [7,106,121]. This cholecalciferol feature may support the generation of 25(OH)D from tissue stores of cholecalciferol [7].

The conversion rate of cholecalciferol into 25(OH)D follows a non-linear pattern, resulting in a plasma 25(OH)D concentration curve that reaches a true plateau at approximately 30–50 ng/mL [106,114,122,123,124,125,126,127]. That is, a lower conversion rate is observed as a result of subsequent D_3_ intake or increasing doses, once 25(OH)D concentrations approach to a certain threshold [114,122,123,125,126,127]. This also correlates with a greater increase (steeper curve) in serum 25(OH)D following cholecalciferol in case of more severe VDD, compared with lower delta 25(OH)D observed in insufficient or even vitamin D-replete patients [127]. Feedback inhibition of enzyme activity at adequate 25(OH)D amounts or intrinsic kinetic features of 25-hydroxylase have been proposed as potential mechanisms for regulation of the cholecalciferol-to-25(OH)D hepatic conversion [109,127]. This pharmacokinetic profile also avoids 25(OH)D fluctuations in serum following individual administrations, otherwise achieving sustained 25(OH)D concentrations [124], which was ideally suggested elsewhere [7].

Altogether, the hepatic hydroxylation step, along with the lack of a linear relationship in the 25(OH)D production, may prevent an indefinite increase in serum values once under treatment, as well as obtain more predictable and stable levels over time at a given target level. In other words, The efficiency of cholecalciferol supplementation in vitamin D-replete patients is physiologically lowered by the organism, possibly to prevent intoxication [128].

Few studies have investigated the pharmacokinetics of vitamin D supplementation in healthy individuals with VDD. In one such study [122], participants were randomized into three groups: (1) cholecalciferol 10,000 IU/day for eight weeks followed by 1000 IU/day for four weeks; (2) 50,000 IU/week for 12 weeks; and (3) 100,000 IU every other week for 12 weeks. The mean baseline serum 25(OH)D concentration across all participants was 13.5 ± 3.7 ng/mL. By week 4 of therapy, 93% of participants had levels above 30 ng/mL, and by day 56, all participants (100%) exceeded this threshold. Serum calcium and phosphate levels increased in all treatment groups by approximately 1.5% and 4.2%, respectively; however, these changes were clinically insignificant. Overall, high doses of vitamin D_3_ supplementation were well tolerated in these healthy participants [122]. Other studies on vitamin D supplementation have reported only modest increases in fibroblast growth factor 23 (FGF23) among individuals with low baseline 25(OH)D concentrations [129,130].

#### 3.1.2. Supplementation Pattern Dosing

Bolus or intermittent doses of vitamin D_3_ are commonly used to rapidly replete VDD in adults and are sometimes prescribed based on patient preference [131]. The effects of various dosing strategies in correcting vitamin D deficiency/insufficiency will be examined, as well as their physiological and clinical impacts, particularly on musculoskeletal health.

Different studies have evaluated the comparative efficacy and safety of vitamin D3 (cholecalciferol) treatment with distinct dosing regimens (daily, weekly, monthly). Mostly, when the same cumulative dose of vitamin D3 is given, normalization of 25(OH)D concentrations over a period of two-three months was found to be similar [110,111,122]. As being daily and monthly dosing regimens equivalent in restoring 25(OH)D levels over time, a more rapid increase was observed in the latter [132]. In the study of De Niet et al. [132], the same cumulative dose of vitamin D_3_ was given as either 2000 IU/d or 50,000 IU/month for 75 days to 60 healthy young adults. The 25(OH)D concentrations in the two groups were similar at baseline (approx. 14 ng/mL), on day 25 (approx. 28 ng/mL), and thereafter; however, the levels differed over the first two weeks of treatment. The monthly dose caused a rapid increase of approximately 9 ng/mL in serum 25(OH)D after only 2 days, whereas the daily dose had increased 25(OH)D by only 2 ng/mL at day 2. A similar pattern was found for 1,25(OH)_2_D. Serum fibroblast growth factor 23 (FGF23) concentrations did not increase in either group. Overall, this study demonstrates that an intermittent 50,000 IU dose of vitamin D can safely and rapidly correct VDD, showing comparable safety and efficacy to the daily dosing regimen. However, other studies have shown a slight superiority with daily administration respect to intermittent patterns [122,133].

Another topic to consider is that cholecalciferol, unlike other molecules such as calcifediol, has demonstrated therapeutic equivalence when administered in daily, weekly, and monthly doses in its corresponding equivalents [110,111]. This is a very important advantage regarding the efficiency of treatments, as monthly doses have shown better long-term patient preference and adherence compared to daily and weekly doses [134,135].

On the other hand, larger bolus dosing must be undertaken with care to avoid adverse musculoskeletal consequences since bolus doses of vitamin D can increase circulating levels of FGF23 [130]. FGF23 downregulates 1α-hydroxylase (CYP27B1) and upregulates 24-hydroxylase (CYP24A1), with the net effect of inadequate 1,25(OH)2D despite adequate or high levels of 25(OH)D [136]. Degradation of 1,25(OH)2D results in decreased calcium absorption, increased bone turnover, bone loss and fractures. It appears that larger bolus doses of vitamin D have increased risk of falls and fractures [116,137,138].

The International Osteoporosis Foundation recommends vitamin D supplementation for individuals aged 60 years or older, with vitamin D3 doses of 800 to 1000 IU per day to benefit bone health and help reduce the risk of falls. Generally, supplementation with 2000 IU of vitamin D_3_ is adequate to increase the 25(OH)D levels to normal within a few weeks [100,120].

The concept of vitamin D safety may be evaluated from two perspectives: the supplementation dose used and the target outcomes (25(OH)D levels, symptoms of clinical hypervitaminosis such as hypercalcemia or urinary calcium, falls, or fractures). According to J-shaped and U-shaped associations between 25(OH)D and morbidity and mortality risk, falls and fractures, levels over 45–60 ng/mL could be potentially harmful [87,116,139]. In this sense, optimal levels are being mostly set in an interval of 30–50 ng/mL [35,54,64,100,105,139].

In addition, the upper tolerable intake level for vitamin D in the general population is set at 4000 IU daily by both IOM and EFSA [87,140]. An upper threshold of supplementation with 10,000 IU/day was established as no observed adverse effect level [53,87,140,141], being considered safe in the long-term [141,142]. However, increased trends of hypercalciuria seem to be elicited with the long-term use of this dose [143].

Regarding clinical outcomes, infrequent, single high-doses boluses (300,000–500,000 IU) have shown to increase the risk of falls or fractures. Recent data have also suggested that long-term administration (1–5 years) of doses of 60,000 to 300,000 IU monthly (or in its daily equivalents) may have also detrimental effects on bone-related parameters [105,116,144,145]. This cutoff is still controversial, as other authors who administered doses of 80,000–100,000 IU monthly did not report any increase on fractures, falls and other adverse events [105,143,146,147]. Altogether, the use of chronic doses of up to 2000 IU/day of cholecalciferol undoubtedly remains in the safety margin [142,148].

Lastly, exogenous vitamin D toxicity (VDT) is characterized by hypercalcemia, clinical hypercalciuria and very low levels of PTH, and most often manifests with clinical symptoms such as confusion, apathy, recurrent vomiting, abdominal pain, polyuria, polydipsia, and dehydration. Vitamin D intoxication is extremely rare and is mostly due to the use of vitamin D metabolites [149], or improper or inadvertent ingestion of very high amounts of vitamin D for a prolonged period of time [39,141]. Existing clinical knowledge on VDT relies on anecdotal case reports, most of which reported or exceeded a continued daily intake above the EFSA upper level limit recommendation or were related to inappropriate use or labeling errors [141,148]. 25(OH)D concentrations over 100 and 150 ng/mL are the hallmarks of hypervitaminosis and vitamin D toxicity, respectively [53,66,105,106,142]. Of note, given the potential for hypercalcemia and clinical hypercalciuria to produce clinical adverse effects, these endpoints should also be critical for the evaluation of toxicity. Most studies involving co-supplementation of calcium and vitamin D_3_ have not reported significant increases in serum calcium or renal stone formation either [141].

### 3.2. Recommendations for Vitamin D Monitoring Needs

Measurements of serum 25(OH)D are not always available, even for vitamin D-deficient risk groups. As previously provided by other clinical guidelines [49,91,118,150,151], recommendations for treated patients are given for those cases.

#### 3.2.1. General Population

Based on the clinical evidence reviewed, it would not be necessary for clinicians to routinely monitor 25(OH)D for safety or efficacy in the general population when supplementation is within the recommended limits [86,91]. In adults, considering the determinants of vitamin D response, appropriate management should include oral supplementation with cholecalciferol (vitamin D_3_) at a dose of 4000 IU/day—or the equivalent weekly or monthly regimen—for initial repletion. For long-term maintenance (>6 months), daily doses of up to 2000 IU may be considered [66,144].

#### 3.2.2. Special Patients

Monitoring should be considered and may be advisable in the following settings to guide dose adjustment and ensure therapeutic efficacy (Table 2).

If vitamin D monitoring is considered necessary, serum 25(OH)D is the preferred metabolite to monitor 25(OH) concentrations [53]. Serum and urinary calcium should also be assessed [116]; Clinicians are advised to test after 3–4 months of initiating vitamin D_3_ supplementation; and following 6–12 months once appropriate levels have been achieved [66].

## 4. Unmet Needs

### 4.1. Implications for Meta-Analyses and Guideline Development

Despite annual publication of over 200 vitamin D systematic reviews, controversy persists regarding the definition of VDD [156]. This controversy is due, in part, to reporting of non-standardized and therefore variable 25(OH)D results, and ensuing variability relating them to major bone health outcomes [78].

The benefit of vitamin D supplementation on fracture prevention has been extensively assessed in several systematic reviews/meta-analyses, reporting discordant conclusions [157,158]. Similarly, more than 40 international vitamin D guidelines have also been identified in two recent SRs, with, as expected, highly variable recommendations [156,159]. On average, each guideline only met 10 (95% CI: 9–12) of the 25 methodological criteria for guideline development recommended by the WHO Handbook [160]. Furthermore, only a minority of the studies were based on rigorous systematic reviews and meta-analyses of relevant research. Equally important, many lacked standardization in the 25(OH)D assays employed [159]. Finally, many studies of vitamin D supplementation recruited primarily vitamin D replete subjects in whom benefit of additional intake is, at best, unlikely [161,162]. Thus, even meta-analyses of such studies do not provide insight into the optimal 25(OH)D level or to the definition of hypovitaminosis D. When combined, the above issues explain the wide variations in recommended levels for serum 25(OH)D concentration, spanning from 10 to 40 ng/mL, but also in recommended daily vitamin D intake [159]. A recent systematic review commissioned by the Endocrine Society synthesized evidence from 151 studies and underpins the 2024 Clinical Practice Guideline, attempting to provide a more robust methodological framework for current recommendations [163]. Such lack of consensus has and continues to paralyze development and implementation of unified, evidence-based, vitamin D guidance to advance clinical care and public health policy. Standardization of vitamin D assays is thus a pressing priority.

### 4.2. Call for Action/Standardization

As hypovitaminosis D is highly prevalent worldwide and is associated with adverse bone health outcomes, and possibly other health consequences, the development of evidence-based guidelines and public health policies are essential. To this end, calls for reporting standardized 25(OH)D values have been made [164,165,166]. Reporting standardized 25(OH)D data would allow meaningful pooling of data for systematic reviews thereby contributing to consensus definition of VDD. Indeed, some journals have started requiring the report of standardized 25(OH)D results as a condition for publication [167]. Acknowledging that measurement of other vitamin D metabolites may be forthcoming and consideration of various pathologic conditions are likely important in assessment of vitamin D status, requiring the reporting of standardized 25(OH)D values as a condition for publication represents a pivotal step toward advancing the vitamin D research landscape. Doing so is essential to allow meaningful pooling of results from various studies that link serum 25(OH)D levels to major outcomes, thereby facilitating consensus regarding what truly constitutes VDD, and thus the formulation of trustworthy guidelines.

## 5. Conclusions

This narrative review examines current evidence on the prevalence, clinical implications, and management strategies of VDD, with particular emphasis on the rationale for screening, supplementation practices, and therapeutic targets in bone health. Although awareness of the clinical implications of hypovitaminosis D is increasing, the implementation of screening strategies remains controversial. Universal screening lacks robust evidence regarding its clinical utility and cost-effectiveness, and although targeted testing in specific risk groups has been proposed, this approach also entails limitations due to variability in assay methods, thresholds, and inclusion criteria across clinical guidelines and studies.

In light of this, empirical supplementation emerges as a pragmatic and widely supported strategy in certain high-risk populations, including institutionalized older individuals, patients with malabsorption syndromes, obesity, or osteoporosis, and women during pregnancy or lactation. In these settings, the routine measurement of serum 25(OH)D concentrations prior to treatment does not appear to offer additional clinical benefit, particularly given the favorable safety profile of oral cholecalciferol and the low incidence of toxicity at recommended doses. Nevertheless, monitoring of 25(OH)D concentrations is warranted in specific clinical scenarios, such as in patients receiving high-dose regimens or vitamin D metabolites, those with chronic liver or kidney disease, metabolic bone disorders, or in the presence of pharmacological interactions or conditions predisposing to hypercalcemia or vitamin D intoxication.

In the context of skeletal health, evidence supports the use of a 25(OH)D threshold of 30 ng/mL to define sufficiency, particularly given its association with optimal parathyroid hormone suppression and improved outcomes in fracture prevention. Moreover, adequate vitamin D status is essential to ensure the efficacy and safety of antiresorptive therapies, as insufficient levels have been linked to suboptimal therapeutic response and an increased risk of hypocalcemia. Among the available compounds, such as calcifediol, cholecalciferol remains the preferred option for supplementation, owing to its superior pharmacokinetic properties, sustained elevation of serum 25(OH)D concentrations, and the flexibility of dosing regimens that favor adherence. Its physiological profile and clinical performance render it particularly suitable for long-term strategies in the prevention and management of VDD.

Future research should aim to refine screening criteria, optimize supplementation strategies across diverse populations, and clarify the long-term clinical outcomes associated with maintaining adequate 25(OH)D concentrations, particularly in relation to bone health and beyond.

## Figures and Tables

**Table 1 nutrients-17-03573-t001:** Selected categories of subjects who require vitamin D supplementation (without a prior vitamin D assay).

Target Population
Subjects with limited sun exposure and at wintertime [6]
Subjects with insufficient vitamin D intake [6]
individuals with pigmented skin [93]
Children and adolescents [6]
Women planning pregnancy, pregnant and breastfeeding women [94]
Older adults (≥65 years), particularly if at risk of fractures [95]
Institutionalized individuals [96]
Subjects at risk of osteoporosis or patients diagnosed with osteoporosis [89,97]: a. On osteoporosis treatment b. With fragility fractures
Obese patients and prior/after bariatric surgery [6]
Patients with malabsorption [6]
Patients with documented hypovitaminosis D [96]: Maintenance dose after vitamin D deficiency treatment.
Patients with high risk of prediabetes or diabetes [6]
Patients with liver diseases: cirrhosis, fatty liver

**Table 2 nutrients-17-03573-t002:** Selected categories of subjects who require serum 25(OH)D monitoring.

Monitoring Indication
Symptomatic vitamin D deficiency [91]
Use of metabolites other than cholecalciferol (e.g., calcifediol, calcitriol) [7]
Supplementation at higher doses other than those mentioned above [66]
Individuals taking medications that interfere with vitamin D absorption or metabolism (e.g., cholestyramine, antiepileptic drugs/inducers of the cytochrome P450 pathway such as phenytoin and phenobarbital, corticosteroids, statins, antimicrobials -imidazole antifungal, actinomycin, rifampicin, hydroxychloroquine, highly active antiretroviral agents-, immunosuppressive and chemotherapeutic agents or antihistamines) [66]
Individuals taking medications known to interact with vitamin D treatment or to induce side effects (e.g., cardiac glycosides, thiazides) [105,149]
patients with anticipated poor adherence [66]
Previous history of hypervitaminosis D, hypo/hypercalcemia or hypercalciuria, and hyperphosphatemia [152]
Malabsorption syndromes and bariatric surgery [153]
Obesity (body mass index > 30 kg/m^2^) [66]
Chronic liver and renal diseases [154]
Metabolic bone diseases, particularly patients under osteoporosis treatments. Previous history of falls and fractures [91]
Hyperparathyroidism and hyperthyroidism [155]
Patients hypersensitive to vitamin D such as 24-hydroxylase deficiency, granuloma-forming disorders, lymphomas, idiopathic intracranial hypertension or Williams-Beuren syndrome [149]

## Data Availability

Not applicable.

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
