# Peer review of "Rethinking Vitamin D Deficiency: Controversies and Practical Guidance for Clinical Management"

_nutrients, 2025, doi:10.3390/nu17223573_

Round 1
Reviewer 1 Report
Comments and Suggestions for Authors
The manuscript provides a narrative (what is this exactly?) review discussing vitamin D deficiency, in turn measured by 25(OH)D status, and its association with various health outcomes.
The question if vitamin D deficiency can be considered causative for onset or progression of diseases, particularly beyond bone diseases, is of enormous interest as well for diagnosis as for the development of therapeutic rationales.
In general, VDD is regarded as a threshold of < 30 ng/mL of 25(OH)D concentration in serum, determined by common immunological assays, that appears too reductionist. In fact, more advanced assays, based on LC-MS/MS, what at least should have to be mentioned in the review, are the gold standard for reliable determination of VDD, that allow for the simultaneous measurement of different vitamin D metabolites with high precision and specificity. Additionally, MS methods can distinguish between VD3 and VD2 metabolites, as well as their 3-epi-derivatives, where immunological assays all fail. Particularly, as relevant criteria for functional VDD, the ratio (VMR) of 24,25(OH)2VD to 25(OH)VD (both D2 and D3) have to be considered (DESIRE-study; Zelzer S, Pilz S,et al, Nutrients 2024,14,16(6):839. VDD means by this 24,25(OH)2VD < 3 ng/mL and VMR <4%. Consequently, modern VD assays, based on LC-MS, (commercially recently launched by ROCHE Diagnostics), the following metabolites have to be measured for differentiated VD diagnostics: 25(OH)VD3, 25(OH)VD2, 24,25(OH)2VD3, 24,25(OH)2VD2, and ideally also 1,25(OH)2VD3, 1,25(OH)2VD2. Finally, functional VDD is a better predictor of morbidity and mortality than just low 25(OH)D concentration (see line 577, 592).
The thresholds of intoxication (i.e. causing hypercalcemia) by supplementation of VD3 and VD2 (Veith et al.), along with their metabolites, particularly 1,25(OH)2VD3, should be described in more detail.
Finally, I may suggest the following minor corrections
Line
26: “attention” instead of “range”
61-64: rephrase
95-104: level of VD is mixed up mentioned in ng/mL or nmol/L, what appears somewhat confusing and difficult to compare. It would be helpful to be consistent (ng/mL or mmol/L)
121: Disorders instead conditions
145: falls…double space
232 < 30 ng/mL
243: individuals instead subjects
266: It
298: Individuals
305: “can be significantly reduced can be significantly prevented or reduced by vitamin D supplementation”: rephrase such as: “can be significantly reduced or even prevented by vitamin D supplementation”
538: Patients

Author Response
Suggestions
In general, VDD is regarded as a threshold of < 30 ng/mL of 25(OH)D concentration in serum, determined by common immunological assays, that appears too reductionist. In fact, more advanced assays, based on LC-MS/MS, what at least should have to be mentioned in the review, are the gold standard for reliable determination of VDD, that allow for the simultaneous measurement of different vitamin D metabolites with high precision and specificity.
Additionally, MS methods can distinguish between VD3 and VD2 metabolites, as well as their 3-epi-derivatives, where immunological assays all fail. Particularly, as relevant criteria for functional VDD, the ratio (VMR) of 24,25(OH)2VD to 25(OH)VD (both D2 and D3) have to be considered (DESIRE-study; Zelzer S, Pilz S,et al, Nutrients 2024,14,16(6):839. VDD means by this 24,25(OH)2VD < 3 ng/mL and VMR <4%.
Additionally, MS methods can distinguish between VD3 and VD2 metabolites, as well as their 3-epi-derivatives, where immunological assays all fail. Particularly, as relevant criteria for functional VDD, the ratio (VMR) of 24,25(OH)2VD to 25(OH)VD (both D2 and D3) have to be considered (DESIRE-study; Zelzer S, Pilz S,et al, Nutrients 2024,14,16(6):839. VDD means by this 24,25(OH)2VD < 3 ng/mL and VMR <4%.
We have added the complete suggestion to the manuscript. Please see lines 134-137 and added a new reference, please see lines 982 and 983.
Consequently, modern VD assays, based on LC-MS, (commercially recently launched by ROCHE Diagnostics), the following metabolites have to be measured for differentiated VD diagnostics: 25(OH)VD3, 25(OH)VD2, 24,25(OH)2VD3, 24,25(OH)2VD2, and ideally also 1,25(OH)2VD3, 1,25(OH)2VD2.
Finally, functional VDD is a better predictor of morbidity and mortality than just low 25(OH)D concentration (see line 577, 592).
We have added this suggestion. Please see lines 356-357.
All the minor corrections have been corrected.
Reviewer 2 Report
Comments and Suggestions for Authors
Please rewrite it. Use more strict statements, change the title of your work, example: Guidelines for prevention and treatment of musculo-skeletal disorders/diseases for Spanish population.
Please correct the whole paper body for "vitamin D levels", it should be 25(OH)D concetration(s), unless you have measure cholecalciferol concentration or ergocalciferol concentration.
Finally, it is lack of any strict statements: how much, how long, what to use, in what conditions...etc.
English is ok.
regards,
Reviewer
Author Response
- Please rewrite it. Use more strict statements, change the title of your work, example: Guidelines for prevention and treatment of musculo-skeletal disorders/diseases population”
Thanks for the reviewers' suggestions. We have different opinions on this, because the review has been carried out using all the existing bibliography, obtaining data available from all countries and not only from the Spanish population. For this reason, we kindly request that we be allowed to keep the original title.
- Please correct the whole paper body for "vitamin D levels", it should be 25(OH)D concentration(s), unless you have measure cholecalciferol concentration or ergocalciferol concentration.
We have done it throughout the whole manuscript.
Reviewer 3 Report
Comments and Suggestions for Authors
- Vitamin D supplementation for dark-skinned and high-latitude populations
I dispute the authors’ recommendation that “pigmented skin people” (p. 7) and people in “areas with scarce sunlight” (p. 4) should be targeted for vitamin D supplementation even if they show no symptoms of vitamin D deficiency.
This is curious because the authors themselves accept that natural selection has acted differently on different human populations. They note that some populations have become darkly pigmented to block UV radiation. As a result, these populations have less vitamin D synthesis in their skin and lower vitamin D levels in their blood. The authors also note that vitamin D levels are lower in areas of scarce sunlight.
If natural selection can modify skin pigmentation, it can also modify vitamin D metabolism. If a population consistently has low vitamin D levels, natural selection will favor those individuals who use vitamin D more efficiently. Eventually, the entire population will be composed of such individuals, and its requirements for vitamin D will be lower.
Adaptation to low levels of vitamin D has been shown by several studies of high-latitude populations, mainly the Inuit of Canada and Greenland (Frost, 2012; Frost, 2018). These populations have adapted physiologically to vitamin D scarcity:
- They absorb calcium at a higher rate from food passing through the intestines (Sellers et al., 2003; also see Rejnmark et al., 2004; Skjøth et al., 2025; Waiters et al., 1999).
- They convert vitamin D to its most active form at a higher rate (i.e., from 25(OH)D to 1,25(OH)2D) (Rejnmark et al., 2004).
- They have carrier proteins that bind more strongly to vitamin D in their bloodstream (Larcombe et al., 2012; Malyarchuk, 2020).
They have also adapted culturally:
- They consume meat only in a raw or boiled state, thus preserving a co-factor that reduces the risk of rickets independently of the meat’s vitamin D content (Dunnigan et al., 2005; Mellanby, 1918).
- Their children are breastfed for at least two years after birth. Mother’s milk is rich in beta-casein and other co-factors that increase the bioavailability of calcium (Frost, 2022; Kent et al., 2009; Lönnerdal, 2003).
A similar adaptation has been shown in dark-skinned populations, specifically African Americans. With each 10% increase in African ancestry, the serum 25(OH)D level decreases by 2.5 to 2.75 nmol/L (Signorello et al., 2010). Levels are lower than 50 nmol/L during the winter in 53 to 76% of African Americans who live in the southern states (Harris, 2006). Yet few African Americans show symptoms of vitamin D deficiency. They have “a lower prevalence of osteoporosis, a lower incidence of fractures and a higher bone mineral density than white Americans, who generally exhibit a much more favorable vitamin D status” (Robins, 2009). Among women 65 years of age, the risk of a hip fracture by age 80 is only 4% for African Americans versus 11% for European Americans (Barrett-Connor, et al., 2005; Harris, 2006). Among teenage girls, calcium retention, bone formation rates, and calcium absorption efficiency are higher in African Americans than in European Americans (Bryant et al., 2003).
We know that a homeostatic mechanism tends to keep the serum 25(OH)D level at a certain set-point. This set-point seems to vary among human populations, being lower in those that live at high latitudes or have very dark skin. The existence of a homeostatic mechanism is shown by a study of African Americans with varying degrees of African ancestry. Both sunlight and diet were 46% less effective in raising the 25(OH)D levels of participants with high African ancestry, in comparison to those with low African ancestry (Signorello et al., 2010).
- Vitamin D intoxication
I also dispute the authors’ claim that “Vitamin D intoxication is extremely rare” (p. 11). Several studies have found that 25(OH)D levels are beneficial within a relatively narrow range, essentially 40 nmol/L to 100 nmol/L in light-skinned humans from the temperate zone. Levels outside this range are associated with higher mortality:
- The total mortality rate is about 50% higher among men whose 25(OH)D levels are below 46 nmol/L or above 98 nmol/L (Michaëlsson et al., 2010)
- The risk of prostate cancer is significantly higher at levels below 40 nmol/L or above 60 nmol/L (Tuohimaa, 2009; Tuohimaa et al., 2009).
- For endometrial, esophageal, gastric, kidney, non-Hodgkin’s lymphoma, pancreatic, and ovarian cancer, the mortality rate is significantly higher at levels below 45 nmol/L or above 124 nmol/L (Helzlsouer et al., 2020).
- The risk of pancreatic cancer is significantly higher at levels above 100 nmol/L (Stolzenberg-Solomon et al., 2010).
- The risk of cardiovascular disease is significantly higher at levels below 50 nmol/L or above 62.5 nmol/L, and the mortality rate for all causes is significantly higher at levels above 122.5 nmol/L (Davis, 2009).
Impairment of cognition may follow the same U-shaped response curve. In mice, poorer maze performance is associated as much with high 25(OH)D levels as with low ones (Lam et al., 2017). Perhaps most worrisome, other mouse studies have shown a similar response curve for the aging process; premature aging is associated with levels that are either too low or too high (Tuohimaa, 2009; Tuohimaa et al., 2009).
If the beneficial range of levels is lower, the upper bound of that range should also be lower. This is seen with the risk of tuberculosis, which remains low among Africans at progressively lower levels of 25(OH)D, whereas Asians have a progressively higher risk across the same range of levels (Huang et al., 2016). This is also seen with vitamin D supplementation, which fails to reduce bone loss or increase bone turnover in postmenopausal African American women at a pre-treatment level of only 47 nmol/L (Aloia et al., 2005).
If the upper bound of the beneficial range is lower, the threshold of toxicity should also be lower. This is seen with calcified atherosclerotic plaque, which becomes progressively worse among African Americans across a range of 25(OH)D levels that is considered beneficial for European Americans (Freedman et al., 2010).
Individuals with dark skin or from the Arctic may thus be misdiagnosed as vitamin D deficient, wrongly prescribed dietary supplements, and thereby pushed over the threshold of toxicity (Frost, 2012; Frost, 2018; Frost, 2022). Keep in mind that vitamin D is fat-soluble and thus accumulates in the body. Unlike vitamin C, it cannot be removed from the body through urination if too much is ingested.
References
Aloia, J.F., Talwar, S.A., Pollack, S., & Yeh, J. (2005). A randomized controlled trial of vitamin D3 supplementation in African American women. Archives of Internal Medicine, 165, 1618–1623. https://doi.org/10.1001/archinte.165.14.1618
Barrett-Connor, E., Siris, E.S., Wehren, L.E., Miller, P.D., Abbott, T.A., Berger, M.L., Santora, A.C., & Sherwood, L.M. (2005). Osteoporosis and Fracture Risk in Women of Different Ethnic Groups. Journal of Bone and Mineral Research, 20, 185–194. https://doi.org/10.1359/jbmr.041007
Bryant, R.J., Wastney, M.E., Martin, B.R., Wood, O., McCabe, G.P., Morshidi, M., Smith, D.L., Peacock, M., & Weaver, C.M. (2003). Racial differences in bone turnover and calcium metabolism in adolescent females. Journal of Clinical Endocrinology & Metabolism, 88, 1043–1047. https://doi.org/10.1210/jc.2002-021367
Davis, C.D. (2009). Vitamin D and health: Can too much be harmful? American Journal of Lifestyle Medicine, 3, 407–408. https://doi.org/10.1177/1559827609338154
Dunnigan, M.G., Henderson, J.B., Hole, D.J., Mawer, E.B., & Berry, J.L. (2005). Meat Consumption Reduces the Risk of Nutritional Rickets and Osteomalacia. Brit. J. Nutr. 94, 983-991. https://doi.org/10.1079/BJN20051558
Freedman, B.I., Wagenknecht, L.E., Hairston, K.G., Bowden, D.W., Carr, J.J., Hightower, R.C., Gordon, E.J., Xu, J., & Langefeld, C.D. (2010). Divers, J. Vitamin D, adiposity, and calcified atherosclerotic plaque in African-Americans. Journal of Clinical Endocrinology & Metabolism, 95, 1076–1083. https://doi.org/10.1210/jc.2009-1797
Frost, P. (2012). Vitamin D deficiency among northern Native Peoples: a real or apparent problem? International Journal of Circumpolar Health, 71(S2), 18001. https://doi.org/10.3402/IJCH.v71i0.18001
Frost, P. (2018). To supplement or not to supplement: are Inuit getting enough vitamin D? Études Inuit Studies, 40(2), 271-291. https://doi.org/10.7202/1055442ar
Frost P. (2022). The Problem of Vitamin D Scarcity: Cultural and Genetic Solutions by Indigenous Arctic and Tropical Peoples. Nutrients, 14(19), 4071. https://doi.org/10.3390/nu14194071
Harris, S.S. (2006). Vitamin D and African Americans. Journal of Nutrition, 136, 1126–1129. https://doi.org/10.1093/jn/136.4.1126.
Helzlsouer, K.J. (2020). Steering Committee of Vitamin D Pooling Project of Rarer Cancers. Abstract PL04-05: Vitamin D: Panacea or a Pandora’s box for prevention? Cancer Prevention Research, 3, PL04–PL05. https://doi.org/10.1158/1940-6207.PREV-09-PL04-05
Huang, S.-J., Wang, X.-H., Liu, Z.-D., Cao, W.-L., Han, Y., Ma, A.-G., & Xu, S.-F. (2016). Vitamin D deficiency and the risk of tuberculosis: A meta-analysis. Drug Design, Development and Therapy, 11, 91–102. https://doi.org/10.2147/DDDT.S79870
Kent, J.C., Arthur, P.G., Mitoulas, L.R., & Hartmann, P.E. (2009). Why calcium in breastmilk is independent of maternal dietary calcium and vitamin D. Breastfeeding Review, 17, 5-11.
Lam, V., Takechi, R., & Mamo, J.C. (2017). Vitamin D, Cerebrocapillary Integrity and Cognition in Murine Model of Accelerated Ageing. Alzheimer's & Dementia, 13, P1304. https://doi.org/10.1016/j.jalz.2017.06.1990
Larcombe, L., Mookherjee, N., Slater, J., Slivinski, C., Singer, M., Whaley, C., Denechezhe, L., Matyas, S., Turner-Brannen, E., Nickerson, P., & Orr, P. (2012). Vitamin D in a Northern Canadian First Nation Population: Dietary Intake, Serum Concentrations and Functional Gene Polymorphisms. PLoS ONE, 7(11): e49872. https://doi.org/10.1371/journal.pone.0049872
Lönnerdal, B. (2003). Nutritional and Physiologic Significance of Human Milk Proteins. American Journal of Clinical Nutrition, 77, 1537S-1543S. https://doi.org/10.1093/ajcn/77.6.1537S
Malyarchuk, B.A. (2020). Polymorphism of GC gene, encoding vitamin D binding protein, in aboriginal populations of Siberia. Ecological Genetics, 18, 243-250. https://doi.org/10.17816/ecogen18634
Mellanby, E. (1918). The part played by an ‘accessory factor’ in the production of experimental rickets. Proceedings of the Physiological Society. xi–xii.
Michaëlsson, K., Baron, J.A., Snellman, G., Gedeborg, R., Byberg, L., Sundström, J., Berglund, L., Ärnlöv, J., Hellman, P., Blomhoff, R., et al. (2010). Plasma vitamin D and mortality in older men: A community-based prospective cohort study. American Journal of Clinical Nutrition, 92, 841–848. https://doi.org/10.3945/ajcn.2010.29749
Rejnmark, L., Jørgensen, M.E., Pedersen, M.B., Hansen, J.C., Heickendorff, L., Lauridsen, A.L., Mulvad, G., Siggaard, C., Skjoldborg, H., Sørensen, T.B., et al. (2004). Vitamin D insufficiency in Greenlanders on a Westernized fare: ethnic differences in calcitropic hormones between Greenlanders and Danes. Calcified Tissue International, 74, 255–263. https://doi.org/10.1007/s00223-003-0110-9
Robins, A.H. (2009). The evolution of light skin color: Role of vitamin D disputed. American Journal of Physical Anthropology, 139, 447–450. https://doi.org/10.1002/ajpa.21077
Sellers, E.A.C., Sharma, A. & Rodd, C. (2003). Adaptation of Inuit children to a low-calcium diet. Canadian Medical Association Journal, 168(9), 1141–1143.
Signorello, L.B., Williams, S.M., Zheng, W., Smith, J.R., Long, J., Cai, Q., Hargreaves, M.K., Hollis, B.W., & Blot, W.J. (2010). Blood vitamin D levels in relation to genetic estimation of African ancestry. Cancer Epidemiology, Biomarkers & Prevention, 19, 2325–2331. https://doi.org/10.1158/1055-9965.EPI-10-0482
Skjøth, J. B., Hagens, T. M., Fleischer, I., Laursen, M., & Andersen, S. (2025). Bone mineral content among Inuit – a systematic review of data. International Journal of Circumpolar Health, 84(1). https://doi.org/10.1080/22423982.2025.2502249
Stolzenberg-Solomon, R.Z., Jacobs, E.J., Arslan, A.A., Qi, D., Patel, A.V., Helzlsouer, K.J., Weinstein, S.J., McCullough, M.L., Purdue, M.P., Shu, X.O., et al. (2010). Circulating 25-hydroxyvitamin D and risk of pancreatic cancer, Cohort Consortium Vitamin D Pooling Project of Rarer Cancers. Am. J. Epidemiol. 172, 81–93. https://doi.org/10.1093/aje/kwq120
Tuohimaa, P. (2009). Vitamin D and aging. Journal of Steroid Biochemistry and Molecular Biology, 114, 78–84. https://doi.org/10.1016/j.jsbmb.2008.12.020
Tuohimaa, P., Keisala, T., Minasyan, A., Cachat, J., & Kalueff, A. (2009). Vitamin D, nervous system and aging. Psychoneuroendocrinology, 34S, S278–S286. https://doi.org/10.1016/j.psyneuen.2009.07.003
Waiters, B., Godel, J.C., & Basu, T.K. (1999). Perinatal Vitamin D and Calcium Status of Northern Canadian Mothers and their Newborn Infants. Journal of the American College of Nutrition, 18, 122-126. https://doi.org/10.1080/07315724.1999.10718839
Author Response
1. Vitamin D supplementation for dark-skinned and high-latitude populations
I dispute the authors’ recommendation that “pigmented skin people” (p. 7) and people in “areas with scarce sunlight” (p. 4) should be targeted for vitamin D supplementation even if they show no symptoms of vitamin D deficiency.
This is curious because the authors themselves accept that natural selection has acted differently on different human populations. They note that some populations have become darkly pigmented to block UV radiation. As a result, these populations have less vitamin D synthesis in their skin and lower vitamin D levels in their blood. The authors also note that vitamin D levels are lower in areas of scarce sunlight.
If natural selection can modify skin pigmentation, it can also modify vitamin D metabolism. If a population consistently has low vitamin D levels, natural selection will favor those individuals who use vitamin D more efficiently. Eventually, the entire population will be composed of such individuals, and its requirements for vitamin D will be lower.
Adaptation to low levels of vitamin D has been shown by several studies of high-latitude populations, mainly the Inuit of Canada and Greenland (Frost, 2012; Frost, 2018). These populations have adapted physiologically to vitamin D scarcity:
• They absorb calcium at a higher rate from food passing through the intestines (Sellers et al., 2003; also see Rejnmark et al., 2004; Skjøth et al., 2025; Waiters et al., 1999).
• They convert vitamin D to its most active form at a higher rate (i.e., from 25(OH)D to 1,25(OH)2D) (Rejnmark et al., 2004).
• They have carrier proteins that bind more strongly to vitamin D in their bloodstream (Larcombe et al., 2012; Malyarchuk, 2020).
They have also adapted culturally:
• They consume meat only in a raw or boiled state, thus preserving a co-factor that reduces the risk of rickets independently of the meat’s vitamin D content (Dunnigan et al., 2005; Mellanby, 1918).
• Their children are breastfed for at least two years after birth. Mother’s milk is rich in beta-casein and other co-factors that increase the bioavailability of calcium (Frost, 2022; Kent et al., 2009; Lönnerdal, 2003).
A similar adaptation has been shown in dark-skinned populations, specifically African Americans. With each 10% increase in African ancestry, the serum 25(OH)D level decreases by 2.5 to 2.75 nmol/L (Signorello et al., 2010). Levels are lower than 50 nmol/L during the winter in 53 to 76% of African Americans who live in the southern states (Harris, 2006). Yet few African Americans show symptoms of vitamin D deficiency. They have “a lower prevalence of osteoporosis, a lower incidence of fractures and a higher bone mineral density than white Americans, who generally exhibit a much more favorable vitamin D status” (Robins, 2009). Among women 65 years of age, the risk of a hip fracture by age 80 is only 4% for African Americans versus 11% for European Americans (Barrett-Connor, et al., 2005; Harris, 2006). Among teenage girls, calcium retention, bone formation rates, and calcium absorption efficiency are higher in African Americans than in European Americans (Bryant et al., 2003).
We know that a homeostatic mechanism tends to keep the serum 25(OH)D level at a certain set-point. This set-point seems to vary among human populations, being lower in those that live at high latitudes or have very dark skin. The existence of a homeostatic mechanism is shown by a study of African Americans with varying degrees of African ancestry. Both sunlight and diet were 46% less effective in raising the 25(OH)D levels of participants with high African ancestry, in comparison to those with low African ancestry (Signorello et al., 2010).
Our response:
We appreciate the reviewer’s thoughtful and well-documented comments regarding the possible evolutionary adaptations to low vitamin D levels in certain populations. We fully agree that this is a valid and interesting line of evidence, and the references provided are highly relevant.
However, addressing this hypothesis in depth would imply a different conceptual approach and a substantially revised manuscript focused on evolutionary physiology rather than on current clinical and public health criteria. For this reason, we have chosen to acknowledge this perspective in the msnuscript (please sse lines 124-130) while keeping the main focus of our paper on present-day recommendations and guidelines for vitamin D supplementation.
Honestly, I had never received such a well-developed and documented reviewer report, and I sincerely appreciate his/here effort.
Round 2
Reviewer 2 Report
Comments and Suggestions for Authors
lines 124 - 129 provide error statements: what does it mean "of vitamin D even at relatively low serum levels" or what does it mean "to serum 25(OH)D levels" - WHAT MEANS "level" in relation to "concentration". Did you measured serum or plasma cholecalciferol concentration? THIS must be corrected.
line 148 provides citation [169] - should be corrected
line 195 "25(OH)D levels" should be corrected....
line 207, line 218 "25(OH)D levels" should be corrected....
line 227 , line 273, line 310, line 558, line 619, line 637: "25(OH) concentrations " should be corrected....add "D", please
The paragraph "Current Guideline Recommendations " should be rewritten: to avoid bone-centrism please add some more guidelines: Polish, Italian, etc. Furthermore, please add some words on calcifediol as alternative way of prevention or treatment
Table 1 : lack of liver disorders- that group of fatty liver, NAFLD, cirrhosis, for example - should be corrected
line 318 - should be "UVB radiation" instead of UVR - should be corrected
line 332 : "Although there continues to be controversy over what the optimal level of 25(OH)D
is, the optimal level of vitamin D at the bone level" - please correct it, too much level(s)....
line 336: "level" should be corrected
line 387: "between 200 and 2,000 IU/day to achieve a goal of circulating 25(OH)D of at least 50 nmol/L (20 ng /ml)" - really? 200 IU/day? - no, no, no, or you are a white living at equator! MUST be corrected!
line 423, line 437; 445, 464, 473, 478, "25(OH)D levels .." to correct
Author Response
Lines 124 - 129 provide error statements: what does it mean "of vitamin D even at relatively low serum levels" or what does it mean "to serum 25(OH)D levels" - WHAT MEANS "level" in relation to "concentration". Did you measured serum or plasma cholecalciferol concentration? THIS must be corrected.
We have corrected it.
line 148 provides citation [169] - should be corrected. This is the correct reference because it was added by the suggestion of another reviewer.
line 195 "25(OH)D levels" should be corrected....
We have corrected it.
line 207, line 218 "25(OH)D levels" should be corrected....
We have corrected it.
line 227 , line 273, line 310, line 558, line 619, line 637: "25(OH) concentrations " should be corrected....add "D", please
We have corrected it.
The paragraph "Current Guideline Recommendations " should be rewritten: to avoid bone-centrism please add some more guidelines: Polish, Italian, etc. Furthermore, please add some words on calcifediol as alternative way of prevention or treatment
We appreciate the reviewers' comments, but we have other thoughts on this point. The reference from the National Osteoporosis Foundation, although focused on bone health, is valid and one of the most frequently cited documents. There are many other guidelines besides the Italian and Polish ones suggested by the reviewer, but our review does not attempt to include exhaustive references to all published guidelines.
Furthermore, the role of calcifediol, as well as other vitamin D metabolites such as 1alpha-cholecalciferol, is not within the scope of our review, and there is no consensus on this issue. Calcifediol does not appear in virtually any guideline.
Table 1 : lack of liver disorders- that group of fatty liver, NAFLD, cirrhosis, for example - should be corrected
We have added a line to the table 1
line 318 - should be "UVB radiation" instead of UVR - should be corrected
We have corrected it.
line 332 : "Although there continues to be controversy over what the optimal level of 25(OH)D
is, the optimal level of vitamin D at the bone level" - please correct it, too much level(s)....
We have corrected it.
line 336: "level" should be corrected
We have corrected it.
line 387: "between 200 and 2,000 IU/day to achieve a goal of circulating 25(OH)D of at least 50 nmol/L (20 ng /ml)" - really? 200 IU/day? - no, no, no, or you are a white living at equator! MUST be corrected!
It was a tipographic error that has been corrected
line 423, line 437; 445, 464, 473, 478, "25(OH)D levels .." to correct
We have corrected it.